# The Constellation of Plans: Toward a New Structure of Comprehensive Plans in US Cities

Christine Quattro and Thomas Daniels *

Department of City and Regional Planning, University of Pennsylvania, Philadelphia, PA 19104-6311, USA
* Correspondence: thomasld@design.upenn.edu

**Abstract:** The comprehensive plan is a fundamental planning document to direct growth and change in land use in US cities. This paper investigates whether US cities are structuring comprehensive plans to resemble a 'constellation' of functional, special topic, and neighborhood subplans tied to a central guiding plan. A traditional comprehensive plan features a central plan with few, if any, subplans. We analyzed the comprehensive plans of 39 cities and identified 20 cities that are using the constellation structure with at least four subplans. We then evaluated the quality of the constellation-type plans and surveyed planners in 14 cities to understand how and why they are drafting and implementing constellation-type plans. The responses of planners illustrate how this approach to comprehensive planning allows cities to increase the size and number of functional and area subplans. The constellation structure enables land use planners to add new topics to the central plan in a timely manner leading to more effective land use planning as cities and environments change.

**Keywords:** comprehensive plan; constellation of plans; subplan; land use; neighborhood plan; plan structure





## 1. Introduction

The comprehensive plan, also known as the general plan or master plan, is one of the fundamental documents of the US planning profession. US cities first adopted comprehensive plans in the early 20th century and these plans have continued to evolve ever since [1–6]. The purposes of a comprehensive plan are to: (1) Establish a vision to guide future land development based on population projections, land use needs, land and water suitability, and adequate infrastructure [1,7,8]; (2) Create an inventory of community assets and provide an analysis of the strengths, weaknesses, opportunities, and threats to those assets [2]; (3) Set goals and objectives for economic development, housing, the environment, land use, community facilities, transportation, urban design, and other functional areas that respond to anticipated community needs [4,9]; (4) Include an Action Plan of how to implement the comprehensive plan through zoning, subdivision and land development regulations, capital improvements programs, and design ordinances [8]; and (5) Provide the legal basis for zoning and land development and subdivision regulations and offer guidance for the capital improvements program to support orderly growth and maintenance [10].

Researchers have focused mainly on plan content and quality rather than the structure of the comprehensive plan, leaving an important gap in the literature [1–20]. Only a few studies have looked at the connectivity between plans; specifically, how functional subplans and area plans are linked to the comprehensive plan and the environmental resilience of the plans [13,14].

To address this gap in the literature, we sought to answer three questions: (1) Are US cities using the constellation structure for their comprehensive plans?; if so, (2) How are cities drafting and implementing the constellation structure and what is the content and

quality of the plans?; and (3) What are the primary strengths and weaknesses of using a constellation-type comprehensive planning model?

The structure and content of the constellation model differs from the traditional comprehensive plan in several important ways. The traditional comprehensive plan features a single plan that is limited to functional chapters—demographics, economic development, transportation, housing, and other physical aspects—typically without in-depth functional subplans or with only a few subplans [1,8,21] (See Figure 1). US planners have used the traditional comprehensive plan primarily to guide zoning, subdivision and land development regulations, and capital improvements for infrastructure [6].

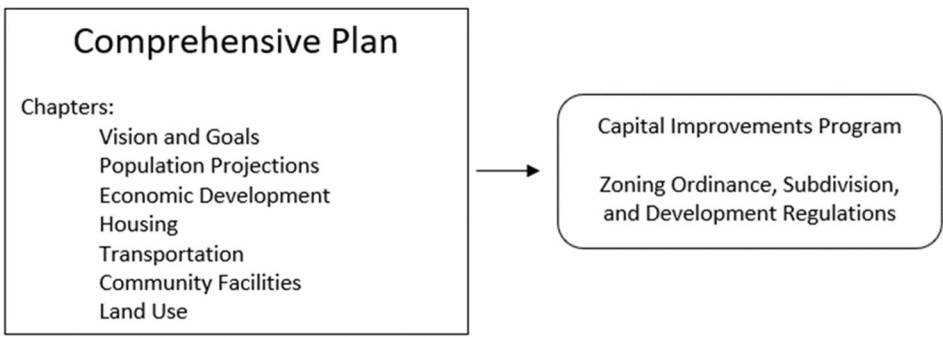

**Figure 1.** The traditional comprehensive plan model.

Through the comprehensive plan, city planners can also direct action on new and anticipated planning needs, including climate change, natural disasters, technological innovations, economic and demographic changes, public health concerns, affordable housing needs, public participation, and social justice [15,16]. This growing number of topics poses a challenge to planners as they attempt to create a timely and robust comprehensive plan. Thus, how are city planners structuring the comprehensive plan to integrate the information and recommendations from these additional plans?

Comprehensive plans may vary in how they connect, coordinate, and integrate functional subplans (such as a transportation plan), special topic plans (climate action plan) and small area subplans (neighborhood redevelopment plan). The structure of a comprehensive plan matters because it can affect plan content, plan quality, timeliness, and actions to respond to new planning issues. The traditional comprehensive planning structure can expand in size and scope as new functional areas and neighborhod plans are added [11,12]. Regional agencies and state planning requirements vary in how they influence the content and procedures for a comprehensive plan and how often the plan must be updated, such as every 10 years, or not at all [6,10]. Such requirements may especially hinder the timeliness of the traditional plan and result in an overly long comprehensive plan document that is not user-friendly.

An alternative structure features a network or "constellation" of functional, special topic, and neighborhood subplans tied to a central comprehensive plan [13,14] (see Figure 2). Figure 2 illustrates how several functional plans and area plans can inform a central comprehensive plan. The functional plans together inform the goals and objectives of the city's comprehensive plan and the recommended actions, such as infrastructure investment and land use regulations. This is a new model of comprehensive planning, incorporating a broader number of topics and in-depth studies compared with the traditional comprehensive plan.

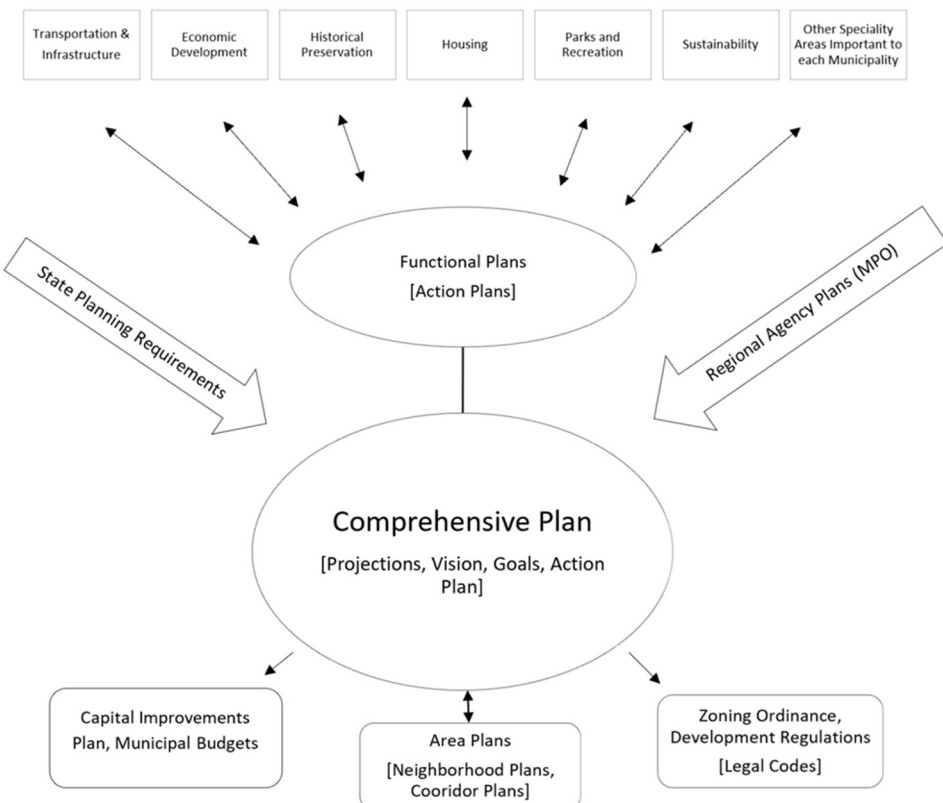

**Figure 2.** The constellation of plans: comprehensive plan model.

The differences between the traditional and constellation plan structures depend on: (1) the number of subplans tied to the central comprehensive plan; and (2) how well the subplans are connected to and inform the comprehensive plan, and vice versa. The constellation of plans structure has more subplans than the traditional comprehensive plan, and thus relies more heavily on the consistency among the subplans [13]. Whether the subplans are well connected to the central comprehensive plan will likely vary from plan to plan, regardless of the structure of the comprehensive plan, constellation-type or traditional [14].

Special topic plans, such as a climate action plan, are often not included within the traditional comprehensive plan, and are therefore not tied to the legal foundations that a comprehensive plan provides for land use regulations [10,22]. The constellation approach enables planners to integrate special topic plans into the comprehensive plan so they can legally influence land use regulations, such as longer required setbacks from streambanks for new buildings in this era of more frequent storms and flooding.

Ideally, the functional subplans inform the comprehensive plan. Many traditional comprehensive plans, however, may lack an in-depth functional subplan (e.g., a Housing Plan) to inform a functional topic (e.g., the housing section of the comprehensive plan). Two potential problems are that the functional subplan, if it does exist, is not well connected to the traditional plan or planners use the traditional plan to set policy for the subplan rather than the other way around [12]. A traditional city comprehensive plan can incorporate functional subplans and neighborhood plans in a single document, but this tends to make the traditional plan very long; this is known as the encyclopedia problem which can overwhelm non-planners who attempt to navigate the plan [23].

The US Planning Enabling Act of 1928, produced by the federal government, provided a model for local governments to create a comprehensive plan, and had this to say about the structure of the plan:

"while the comprehensive or master plan should be envisaged and treated as an organic single unity or whole, there may be no imperative necessity for withholding

the completion and publication of parts as they are finished to await the conclusion and publication of the whole. . . . . . By part may be meant a territorial part. . . . . . "Part" may also relate to subject matter" [19,20,24].

The federal model preferred the comprehensive plan as a single dynamic document, but did not rule out supporting documents, such as neighborhood plans, functional subplans, and strategic plans that inform and further the goals, objectives, and recommended actions of the comprehensive plan. In other words, a comprehensive plan could be structured as a constellation of subplans with the central comprehensive plan as the unifying document. In addition, the subplans could be in a constant or near-constant state of upgrading to better inform the comprehensive plan, creating a more up to date comprehensive plan, and hence a plan that is more responsive to new conditions and priorities, such as adapting to climate change.

In the constellation structure, the subplans are drafted as separate documents but should be linked to the comprehensive plan in two main ways: (1) written statements in both the subplan and the comprehensive plan that relate to each other; and (2) the comprehensive plan makes reference to the subplans and presents a synthesis of the subplans to inform the goals and functional chapters of the comprehensive plan, such as an economic development subplan that informs the economic base chapter.

An important decision is how to integrate functional topics into the comprehensive plan and which functional topics to include. The planning literature is helpful here. Functional topics, such as housing and transportation, typically comprise chapters of any comprehensive plan [4,8]. Separate in-depth functional subplans, such as a housing plan or transportation plan, can inform the respective chapters in the comprehensive plan [ibid.]. We contend that the use of functional subplans and neighborhood plans is more likely in the constellation structure. In other words, if a traditional comprehensive plan makes use of several functional subplans, then the traditional comprehensive plan is really a constellation-type of plan [12,13]. A traditional comprehensive plan will use few, if any subplans, and may not be able to use them effectively. For example, Redaelli [14] discusses how Portland, Oregon struggled with limited success to integrate historic preservation into its traditional comprehensive plan. Although neighborhood plans were part of the comprehensive plan, the historic district subplans did not adequately inform the comprehensive plan and vice versa.

The content of the traditional comprehensive plan tends to favor physical functional elements and economic development and is often less responsive to the changing social and environmental needs of the city [7,25]. Adding new sections to the comprehensive plan typically occurs only during a re-write of the primary comprehensive plan, usually about every 8 to 10 years, but often longer. The constellation of plans may give city planners the opportunity to use the comprehensive plan to coordinate a broad array of subplans to add depth to functional and neighborhood sections of the comprehensive plan and to incorporate elements of special topics plans that address social and environmental aspects of planning. In short, in the constellation structure, planners may be able to add subplans quicker as needs arise. In interviews with planners using the constellation structure, we explored these issues.

Recent research on comprehensive plans has focused mainly on expanding the content of the plans and evaluations of the quality of plans [14,15,18,23–26]. The broader content of plans features a greater focus on sustainability, climate change, community health, smart growth, and equity [10,12–15,25–31]. However, to address a greater number of planning issues, many local governments have adopted special topics plans that fall outside of the traditional comprehensive plan. Such plans include climate action plans, hazard mitigation plans, sustainability plans, equity plans, as well as the more common in-depth functional subplans—bike and pedestrian plans, parks and recreation plans, stormwater management plans, and energy plans, among others [8,32–39]. There has also been discussion on the need cities to develop policies that foster resilience and quickly respond to changing community needs [25–27]. In most cases, it is unlikely that these discrete special topics

plans and functional subplans and their recommendations provide a legal basis for land use regulations, or a way to change those regulations. Tying stand-alone special topics plans and functional subplans to the comprehensive plan gives legal foundation to those discrete plans and subplans as they influence the goals, objectives, and recommended actions expressed in the central comprehensive plan.

Lyles and Stevens [20] reviewed more than 45 articles on plan quality evaluation and found that there are "multiple approaches to plan evaluation" (p. 433). Based on our analysis of US comprehensive planning documents and the survey feedback from city planning departments, we suggest that a good quality constellation-style of comprehensive plan offers several advantages over the traditional comprehensive plan. The constellation-type comprehensive plan can enable: (1) greater consistency between the central comprehensive plan and the discrete functional, special topics, and neighborhood plans; (2) the crafting of zoning and subdivision regulations to implement recommendations originating from functional, neighborhood, and special topics subplans; (3) better direction of infrastructure investments; (4) a greater ability to achieve the goals and objectives of what had been stand-alone special topics plans; (5) planners, planning commission members, and elected officials to refer to a wide array of subplans in making day-to-day planning decisions; (6) a greater timeliness in updating and implementing the comprehensive plan, including the ability to add subplans and adapt the comprehensive plan to enhance community resilience and sustainability; and (7) The ability for specialized drafting of some sub-plans as needed, thus incorporating more detailed technical elements.

In summary, we wanted to show that there is an on-going evolution in the structure of comprehensive plans from a traditional single document to a constellation of plans tied to a central plan, and how cities are using the constellation structure to address a greater array of planning issues, provide a greater depth of information, and even create more effective plans and better outcomes on the ground.

## 2. Materials and Methods

To investigate whether cities are using the constellation structure, we reviewed the comprehensive plans of 39 US cities in a random stratified sample, ensuring a diversity of cities across several variables: geographic location, population size and density, land area, and cities in both isolated metro areas and mega-regions [31]. The selected cities either were currently working on a new comprehensive plan or had a plan less than 10 years old.

Nine cities were in the West, sixteen in the South, seven in the Northeast, and seven in the Midwest. The population of the cities ranged from 46,000 to more than 3,000,000, and the population density varied from fewer than 2000 people per square mile to more than 11,000. Land area included cities of 20 square miles and more than 400 square miles. Eleven cities were in isolated metro areas, and twenty-seven cities were part of a mega-region of two or more major cities.

Next, we analyzed the 39 comprehensive plans to determine which cities were using the constellation model. We deemed cities to be using a constellation-type of comprehensive plan if:

(1) The city had a central comprehensive plan and unctional, special topics, or area subplans which were formatted and presented as separate documents:

    (a) The constellation had subplans that provided in-depth information and analysis for functional elements in the central comprehensive plan. We looked for cities with a comprehensive plan that addressed specific functional topics (housing, land use, transportation, etc.) that drew from distinct subplans

    (b) The subplans were often adopted separately and at a different time than the central comprehensive plan and were drafted by authors who did not write the comprehensive plan;

(2) The city's comprehensive plan was verbally linked to these subplans and incorporated them into its goals and objectives; and

(3) The comprehensive plan contained from four to 20 separate subplans linked to a central comprehensive plan in a constellation structure. Thus, if a comprehensive plan had fewer than four subplans, we did not consider it to have a constellation-type of structure.

Twenty cities met our selection criteria for the constellation of plans model (see Table 1). We then conducted an analysis of plan quality for these 20 cities [20,28–31]. Previous studies have used a scorecard approach to rate the quality of comprehensive plans for specific categories, such as health [11,30], and resilience [14]. We evaluated the quality of the 20 plans in three categories—Plan Connections, Plan Consistency, and Plan Bridging Gaps—and came up with an overall score.

**Table 1.** 20 Cities identified as using a constellation of plans comprehensive plan model.

| State | City (Cities in Bold Responded to the Survey) | Comp Plan Date | Projection To | Population ** (2020) | Land Area (sq.miles) *** | Population Density (per sq. mi) (2020) | Region of US | Isolated Metro * |
|---|---|---|---|---|---|---|---|---|
| AZ | **Tucson** | 2013 | 2023 | 545,340 | 226.7 | 2343 | West | Y |
| CA | **Los Angeles** | 2015 | 2035 | 3,898,747 | 502 | 8485 | West | Y |
| CA | **Sacramento** | 2019–2020 | 2035–2040 | 503,482 | 99.77 | 5242 | West | N |
| CO | **Denver** | 2019 | 2040 | 715,878 | 154.7 | 4674 | West | Y |
| CT | New Haven | 2015 | 2025 | 130,381 | 20.14 | 7170 | Northeast | N |
| FL | **Orlando** | 2008–2017 | 2030 | 284,817 | 119.1 | 2641.70 | South | N |
| IL | **Chicago** | 2014 | 2040–2050 | 2,746,388 | 234 | 11,750.30 | Midwest | N |
| LA | Baton Rouge | 2018 | 2038 | 222,191 | 88.52 | 2982.5 | South | N |
| MA | Baltimore | 2006–2018 | 2016–2022 | 602,274 | 92.28 | 7428 | Northeast | N |
| MS | **Biloxi** | 2009–2020 | 2045 | 46,042 | 67.83 | 1073 | South | N |
| NE | **Omaha** | 2010–2021 | 2030 | 479,529 | 144.6 | 3660 | Midwest | Y |
| NJ | **Trenton** | 2019 | 2042 | 83,387 | 8.205 | 11,102 | Northeast | N |
| NY | Syracuse | 2014 | 2040 | 142,553 | 25.57 | 5636.3 | Northeast | Y |
| OR | Portland | 2018 | 2035 | 650,380 | 145 | 4890.59 | West | Y |
| SC | **Charleston** | 2018 | | 137,041 | 156.6 | 1198.69 | South | Y |
| TX | San Antonio | 2015–2016 | 2040 | 1,598,960 | 505 | 2875.86 | South | N |
| TX | **Austin** | 2012 | 2040 | 965,872 | 271.8 | 3006.4 | South | N |
| UT | **Salt Lake City** | 2015 | 2040 | 200,133 | 110.8 | 1722 | West | Y |
| VA | **Roanoke** | 2019 | 2040 | 99,122 | 42.85 | 2331 | South | Y |
| WA | Seattle | 2017 | 2035 | 741,251 | 83.78 | 7962 | West | Y |

* No other major city within 100 miles. ** Based on 2020 Census Estimates, all case cities except Baton Rouge, Trenton, Syracuse, and Baltimore experienced population from 2010 to 2020. *** June 2022 Land Area. Note: The 19 cities that were not selected include: Montgomery, AL, Phoenix, AZ, Miami, FL, Tampa, FL, Atlanta, GA, Roswell, NM, Honolulu-Oahu, HI, Indianapolis, IN, Kansas City, KS, Louisville, KY, Portland, ME, Charlotte, NC, Raleigh, NC, Cincinnati, OH, Philadelphia, PA, Pittsburgh, PA, Memphis, TN, Fort Worth, TX, Milwaukee, WI.

For the sake of consistency, only one researcher read and reviewed every plan and subplan, whether adopted or published in draft form, to evaluate plan quality: Plan Connections, Plan Consistency, and Plan Bridging Gaps. Each of the 20 plans was rated in the three evaluation categories and the overall plan quality as (++, +, −, - - -). The rating reflected how well the plan met each quality category and led to an overall plan score (see Table 2).

**Table 2.** Evaluations of the 20 city plans that use the constellation structure.

| State | City | Plan Connections | Plan Consistency | Plan Bridging Gaps | Overall Plan Evaluation |
|---|---|---|---|---|---|
| AZ | Tucson | ++ | ++ | ++ | ++ |
| CO | Denver | ++ | ++ | ++ | ++ |
| CT | New Haven | ++ | ++ | ++ | ++ |
| IL | Chicago | ++ | ++ | ++ | ++ |
| LA | Baton Rouge | ++ | ++ | ++ | ++ |
| NE | Omaha | ++ | ++ | ++ | ++ |
| SC | Charleston | ++ | ++ | ++ | ++ |
| TX | Austin | ++ | ++ | ++ | ++ |
| NY | Syracuse | ++ | + | ++ | ++ |
| CA | Los Angeles | ++ | + | + | + |
| WA | Seattle | + | + | ++ | + |
| CA | Sacramento | + | + | ++ | + |
| FL | Orlando | - | + | ++ | + |
| MS | Biloxi | ++ | + | - | + |
| OR | Portland, OR | ++ | + | + | + |
| TX | San Antonio | + | - | ++ | + |
| VA | Roanoke | - | + | ++ | + |
| UT | Salt Lake City | + | - | - | + |
| MA | Baltimore | + | - | – | - |
| NJ | Trenton | - | – | – | - |

Definitions of Categories: **Plan Connections**; The degree to which the central comprehensive plan and the functional and area subplans are connected to one another with explicit referencing. **Plan Consistency**; The degree to which subplans and the central comprehensive plan are consistent with one another, avoiding contradictory objectives or statements. **Plan Bridging Gaps**: The degree to which the plan constellation covers both the neighborhoods and the functional areas of the city, with special attention to areas of importance outlined by the city itself. **Overall Plan Evaluation**: The degree to which the city has executed these three areas, as well as maintained regular updates, input processes, and clear goals, objectives, and plan language. Range: (high) ++, (medium) +, (low) -,–.

The first category, Plan Connections, measures the degree to which the central comprehensive plan and discrete subplans are interrelated and explicitly reference one another in the text. A strong Plan Connections score (++) indicates that each subplan was part of a cohesive integrated constellation of subplans tied to the central comprehensive plan. A weak Plan Connections score (- -) indicates that the subplans had no connection to one another, were often drafted many years apart, and did not tie to a central comprehensive plan, either explicitly or through unified content. In some cases, such as San Antonio, rated (+) there was a cohesive constellation of subplans, but some special topics plans existed outside of the constellation, such as their Climate Action Plan. There were also multiple transportation plans drafted by different entities, only one of which was tied to the comprehensive plan. If only a few subplans were connected to the central comprehensive plan through explicit language or in a shared vison and goals, then the city's plan, such as Orlando's, was rated (rated -).

The second category, Plan Consistency, reflects the degree to which the subplans did not contradict each other or contain divergent goals. A strong Plan Consistency score (++) indicates that the subplans and the central comprehensive plan shared the same mission and goals, and that the strategies to attain those goals were similar. Minor inconsistencies in plans resulted in a (+) rating. However, the rating was reduced if the subplans had major

conflicting priorities. In San Antonio (rated -), for example, there were 3 transportation subplans, each advocating for a very different transit option as the primary transportation goal. A poor Plan Consistency score (- -) indicated that the subplans had contradictory goals either spatially or in financial allocations. A poor score also meant that some of the city's functional subplans were more than 10 years out of date.

The third category, Plan Bridging Gaps, is a measure of how well the comprehensive plan addressed the needs of the entire city and the breadth of functional topics. A strong score (++) indicates that the city had neighborhood subplans for the entire city and functional subplans to support the corresponding functional section in the comprehensive plan. Scores of somewhat positive (+) and somewhat negative (-) denoted plans that had small to moderate plan gaps, such as one or two missing neighborhood plans or a comprehensive plan goal advocating for preservation of important historical resources, but no historic preservation subplan. A poor score (- -) indicated that the city is missing neighborhood subplans for large parts of the city and/or does not have functional subplans relevant to the goals of their comprehensive plan. In several instances the functional subplans were more than 10 years out of date, even though the comprehensive plan had been updated. This resulted in functional subplans that did not inform the goals of the comprehensive plan. conflicting priorities. In San Antonio (rated −), for example, there were 3 transportation subplans each advocating for a very different transit option as the primary transportation goal. A poor Plan Consistency score (- -) indicated that the subplans had contradictory goals either spatially or in financial allocations. A poor score also meant that some of the city's functional subplans were more than 10 years out of date.

The total plan score combined the three plan quality scores and provided a rating of the plan as excellent, proficient, or poor.

To follow up on our analysis of plan quality, we sent a questionnaire via e-mail to the planning staff of the 20 selected cities. We received responses from 14 cities which included a diversity of city sizes and geographic regions, and excellent, proficient, and poor rated constellation-type plans. The respondents were either senior level planners or planning directors.

The questionnaire asked specific questions and requested open-ended comments to gather information about the process of drafting and implementing a constellation-style comprehensive plan. We wanted to understand in more detail: (1) what are the main subplan topics and how are these selected; (2), who drafts each subplan and what role do planners play in plan drafting and to what degree this differed across topic areas; (3) how is the drafting of the subplans coordinated with each other and with the central comprehensive plan; and (4) what are the barriers to implementing the constellation-style comprehensive plan.

## 3. Results

Of the 39 cities we examined, 19 cities had a single document comprehensive plan with fewer than four subplans. The comprehensive plans ranged in age from 2005 to 2020; five comprehensive plans were more than 10 years old; six plans were five to 10 years old; and eight plans were less than 5 years old. Twenty cities had comprehensive plans with a constellation of 4 to 20 subplans linked to a central comprehensive plan. In 14 of the 20 selected cities, the comprehensive plan was less than 5 years old. In the other six cities, either the comprehensive plan was 5 to 10 years old, or a new comprehensive plan was underway, with drafts online and available for review. Thus, generally, the constellation-type of plans were newer than in the cities with traditional comprehensive plans.

We looked at cities based on population, geographic location, and whether they were part of an isolated metro area or a non-isolated area. We found no notable differences among these three features in the 20 cities that use the constellation plan approach and the 19 cities that do not. Among the constellation plan cities: nine were large cities with more than 500,000 residents, four were mid-size cities of 200,000 to 499,999 people, and seven were small cities of less than 200,000 in population. For the cities without constellation plans, ten

were large cities, five mid-sized cities, and three small cities. Of the twenty constellation plan cities, two are in the Midwest, four in the Northeast, seven in the South, and seven are in the West. Of the cities without constellation plans, four are in the Midwest, three in the Northeast, nine in the South, and three in the West. Nine of the cities with constellation plans are in isolated metro areas and eleven are in non-isolated metro areas. Among cities without constellation plans, eight cities are situated in isolated metro areas and eleven cities in non-isolated metro areas. Given these results, it is unclear whether a larger sample size of cities would reveal a trend for adopting the constellation plan approach based on city size, geographic location, and setting of the metro area. Further research over time could further identify whether the trend in cities adopting the constellation plan structure is increasing, at what rate, and where.

We then evaluated the quality of the constellation-type plans. We found that all but two of the twenty selected cities had at least a proficient quality of Plan Connections, Plan Consistency, and Plan Bridging Gaps among their plans (see Overall Plan Evaluation in Table 2). Nine cities have plans that we rated as excellent in connectivity and consistency. Two examples are Tucson and Denver, whose plan constellations were created systematically and cohesively. There is clear consistency across the subplans, and the subplans were completed within a relatively short period of time.

Nine other cities had good plans but had many functional and area subplans which produced some inconsistencies. Seattle, for instance, has 33 different neighborhood subplans in addition to 14 functional subplans, which suggests longer timelines for updating and maintaining subplans as well as the central comprehensive plan. Seattle's many plans also lacked consistency with one another, possibly because different authors were responsible for different plans over time. In our survey, respondents noted that the change in city leadership and planning staff which could take place while drafting and updating subplans was one of the biggest challenges of the constellation-type model.

We found that two cities, Trenton and Baltimore, had low quality constellation- type comprehensive plans. One city had two distinct constellations of plans, rather than a single unifying plan, and the two plans contradicted each other. This could easily cause confusion and conflict in decision making. The other city drafted subplans without a strong central comprehensive plan to synthesize the subplans and create a cohesive vision or action plan. The result was a collection of subplans that were inconsistent with each other, in part because they were drafted by different people at different times and with varying formats and levels of information, leaving some large planning gaps. There were only area subplans for certain portions of the city, and the selection of where and how to plan appeared to be influenced more by immediate needs instead of a unified vision for the entire city.

Fourteen cities had constellation-type plans that we rated as excellent for bridging gaps. Two were rated good (+) and four somewhat poor (−). A main strength of the constellation-type of comprehensive plan is that it enables planners to add functional subplans rather easily. Many of the cities we surveyed had added functional subplans to expand the content and goals of the central comprehensive plan. Plans that rated poorly had some of the oldest functional subplans. One city appeared to re-adopt some functional subplans, perhaps to comply with state law, without making necessary amendments to reflect changes in the goals and conditions of the city. For cities in states with comprehensive planning laws, such as Maryland, or cities using a consulting agency for planning, plans may be drafted using a template for functional subplans which tends to omit planning issues important to that city.

The use of the constellation structure does not require a city to be a certain size or have a particularly well-staffed or well-funded planning department. Two of the cities with excellently rated plans have under 150,000 residents and four cities have more than 500,000. Similarly, our two lowest rated cities vary considerably in population. Even so, smaller cities, such as Trenton, may have limited resources and face challenges compounded by social justice concerns.

*Questionnaire Responses*

Fourteen of the twenty selected cities responded to our questionnaire (see Table 3). The questionnaire was designed to learn about the process of organizing, drafting, and implementing a constellation-style comprehensive plan and to provide insight into why the trend toward constellation-type plans is happening. We wanted to understand in more detail: (1) what are the main subplan topics and how are these selected; (2), who drafts each subplan and what role does the city planning department play; (3) how is the drafting of the subplans coordinated with each other and with the central comprehensive plan; and (4) how is the constellation-style comprehensive plan implemented and what are the barriers?

**Table 3.** Survey responses from 14 cities about their network of plans structure.

| Subplan and Central Comprehensive Plan Description, Drafting and Implementation | | Number and Percentage of Responses | |
|---|---|---|---|
| Most common subplans | • Transportation;<br>• Housing;<br>• Area–Neighborhood plans;<br>• Sustainability. | 13<br>13<br>13<br>11 | 93%<br>93%<br>93%<br>76% |
| Least common subplans | • Cultural/Arts;<br>• Parks;<br>• Annexation/Growth Management. | 1<br>1<br>0 | 7%<br>7%<br>0% |
| Most common subplans drafted by non-planning staff or outside consultants | • Transportation;<br>• Infrastructure/Streets;<br>• Sustainability;<br>• Neighborhood/Small Area. | 11<br>8<br>7<br>7 | 77%<br>57%<br>50%<br>50% |
| Most common subplan drafted by consulting agencies | • Transportation. | 8 | 57% |
| Most common subplans to have been implemented within 5 years | • Transportation;<br>• Area–Neighborhood;<br>• Housing;<br>• Historic Preservation. | 9<br>9<br>7<br>7 | 64%<br>64%<br>50%<br>50% |
| Most common subplans currently being implemented | • Transportation;<br>• Area–Neighborhood;<br>• Sustainability;<br>• Housing;<br>• Historic Preservation;<br>• Land Use;<br>• Streets/Infrastructure. | 12<br>12<br>10<br>11<br>10<br>8<br>8 | 86%<br>86%<br>71%<br>78%<br>71%<br>57%<br>57% |
| Most common subplan currently being drafted | • Area–Neighborhood. | 8 | 57% |
| Cities that | Amend plans outside of regular updates. | 10 10 | 71% |
| | Have subplans written by city staff outside of the planning department. | 4 | 29% |
| Most common causes of delayed comprehensive plan implementation | • Funding;<br>• Politics;<br>• Lack of Planning Staff. | 14<br>12<br>9 | 100%<br>86%<br>64% |

**Table 3.** *Cont.*

| Subplan and Central Comprehensive Plan Description, Drafting and Implementation | | Number and Percentage of Responses | |
| --- | --- | --- | --- |
| Least common causes of delayed comprehensive plan implementation | • Federal or State Restrictions;<br>• Departmental Coordination;<br>• Lack of Land or Adequate Space. | 11<br>9<br>9 | 78%<br>64%<br>64% |
| Cities in which | The comprehensive plan will be fully implemented within 5 years. | 0 | 0% |
| Cities with | Sub-plans in progress (as part of the constellation of plans). | 12 | 86% |
| Cities in which | Drafting the comprehensive plan involved a large public input process. | 10 | 71% |
| | Drafting the comprehensive plan took more than two years. | 11 | 76% |
| | The comprehensive plan was drafted entirely in-house (without the use of a planning consultant). | 7 | 50% |

Respondents reported that transportation, housing, area plans, and sustainability plans were the most common subplans; cultural/arts, parks, and annexation/growth management plans were the least common subplans (see Table 3). In the constellation of plans some or all of the subplans are drafted by different entities (consultants, the planning department, or another city department.

Transportation subplans were the most likely subplans to be drafted by a consult-ant or city staff outside of the planning department. Given the technical knowledge required to draft a transportation plan, engineers and transportation planning con-sultants are often needed.

Sustainability subplans also relied heavily on outside expertise. Transportation, infrastructure, sustainability, and neighborhood subplans were the most likely to be implemented. These results demonstrate a need for planners to expect to work collaboratively with specialized fields related to their plans, and that such collaboration may increase the feasibility of implementation.

One reason why the trend toward the constellation plans is happening now is suggested by the fact that 7 of the 13 cities reported that they had updated their subplans and comprehensive plan outside of their regular timetable for comprehensive plan revisions or the drafting of an entirely new plan. This suggests that the constellation model may be easier to update than the traditional comprehensive plan. One planner commented, 'Different elements of our comprehensive plan are updated at different times. We recently updated our Mobility Element and created a Health Element. We are currently updating our Housing Element.' Another planner explained, 'We update our small area plans often because some are pushing 20 years in age and times have changed and we support those changes with comprehensive plan goals and other policies that are more relevant.' Similarly, a third planner observed, 'There is a state mandate to revise the comp plan every 5 years. We also amend the plan every time we adopt smaller plans into the comprehensive plan.' Thus, the constellation-type of comprehensive plans suggest a more flexible and continuous approach to updating plans than the traditional comprehensive plan.

Another reason for the popularity of the constellation-type of plan appears to be the increased importance of functional subplans. Planners in the cities with the constellation plans report spending considerable time on drafting or revising functional subplans and area subplans. In part because of the substantial cost of creating an entirely new comprehensive plan, functional subplans have become especially important for informing the central comprehensive plan as well as aiding in the hunt for state and federal grants. Respondents

in all 14 cities reported that lack of funding was the main reason for delays in the implementation of the comprehensive plan. To overcome this challenge, planners noted that tailoring specific functional subplans in a constellation format can create a comprehensive plan that is more competitive in acquiring funding. Federal or regional funding agencies that offer grants for city plan implementation, such as Metropolitan Planning Organizations and the federal Department of Housing and Urban Development, have specific data and policy requirements. If a city's planners draft specific subplans with in-depth data and analysis, they may be better able to demonstrate the needs of their city and how their city will spend the money once it is granted. A 2019 study found that strategies and attention to the linking of plans to budgeting was generally lacking in US cities [5]. Our survey results suggest that the constellation approach to comprehensive planning could help to better align a city's plans and infrastructure budgeting process.

Politics was rated the second most common reason for delays in plan implementation. In the United States planning system, elected officials have the sole authority to legally approve land use regulations, such as zoning, and spending for public infrastructure projects. Planners must not only work collaboratively with elected officials but also can actively educate and engage with elected officials and city residents about planning to expedite the planning process. A constellation of plans enables planners to work with elected officials and the public on specific functional topics or neighborhood subplans. This has the potential to engender greater public engagement and understanding of the planning process and increase community support for the functional and neighborhood subplans and their integration into the comprehensive plan. Community meetings can be smaller and more tailored by subject. For example, planners could meet with people who are interested in the historic preservation element when that subplan is being drafted, rather than hold a large meeting that tries to cover every functional topic in the comprehensive plan. Such engagement on specific planning topics could mitigate politics as a hindrance to timely adoption of subplans as well as changes to the central comprehensive plan.

Respondents pointed out that the number of planners in the department was a major factor in the ability to execute, implement, and update their plans. Cities in the United States range widely in population and each city has its own planning department. Some cities may have only a single full-time planner and others may have more than a hundred planning staff. The constellation of plans model encourages a city's planners to focus on the subplans they can complete in-house and outsource those subplans that require additional expertise. This strategy allows multiple subplans to be enacted concurrently without impacting the timeline for each and could result in better quality plans as the best people for the job are acquired for each functional or neighborhood subplan.

None of the responding cities stated that their comprehensive plan would be fully implemented within five years. This correlated with every city involved in the on-going drafting of subplans, which is typical of the constellation-style comprehensive plan. Unlike a single document comprehensive plan, the constellation of plans features a continuous comprehensive planning process that is almost never completed because each subplan has its own timeline for updates or a new draft.

The cities that did not frequently update subplans or comprehensive plans were the smaller cities in our survey. Those cities were also the most likely to have the majority of their plan drafted by external consultants, which made for an extra expense. Any update to a subplan or the central comprehensive plan could prove quite costly. This suggests that it may be more cost effective to use in-house planners whenever possible, because a constellation-type plan involves continuous updates.

Finally, we sought to determine if the constellation-type of plan is gaining traction globally. We selected 11 plans at random in cities outside the US and found 4 using a constellation-type of comprehensive plan (see Table 4). These are Calgary and Winnipeg in Canada, Manchester, UK, and Brisbane, Australia. This suggests that, at least in English-speaking nations, the constellation-type plan is becoming a useful tool to planners.

**Table 4.** International city comprehensive plans.

| City | City |
|---|---|
| Toronto, Canada | Adelaide, Australia |
| Calgary, Canada * | Culiacán, Mexico |
| Winnipeg, Canada * | Barcelona, Spain |
| Manchester, UK * | Madrid, Spain |
| Birmingham, UK | Islamabad, Pakistan |
| Brisbane, Australia * | |

* City with a constellation-type of comprehensive plan.

## 4. Discussion

Based on our analysis of 20 constellation-style city comprehensive plans and the responses of the planners in 14 cities, we identified four main challenges with the constellation model:

(1) The potentially large variety of planning issues addressed in the subplans which requires planners to collect, analyze, and synthesize a substantial amount of data in a timely manner;

(2) Maintaining consistency among the many subplans in the constellation, often because of the different ages of the subplans and because the subplans are drafted by different entities (consultants, planning department, or another city department);

(3) Selecting and prioritizing goals and objectives from the many subplans within the central comprehensive plan; and

(4) The cost of creating and updating large numbers of subplans as well as the central comprehensive plan.

The constellation of plans can easily become encyclopedic in size. While this may increase its usefulness to planners as a resource, it can also deter non-planners from using the subplans. For example, a comprehensive plan is typically available on a city's website. A single pdf file of the city's comprehensive plan is far more accessible than a set of links to subplans, which can confuse and discourage non-planners. The warning of Hammer, Greene, and Siler Associates in their review of the old US Housing and Urban Development-funded 701 plans is also relevant for constellation plans: '[the] predisposition to produce documents has left a void in the development of an appropriate structure for implementation, review, and modification' [40]. In short, a comprehensive plan, whether a single document or connected to a set of subplans, is only useful if it is implemented.

Subplans and the central comprehensive plan must be consistent with one another as we noted in our assessment of the quality of the constellation-type plans. If an Historic Preservation subplan calls for the preservation of buildings from a certain era, but the Economic Development subplan calls for higher density and infill development, this can also be confusing for both public officials and private developers. The subplans may be equally valid, but preference would likely be for the newer subplan. This is a tendency in the 20 constellation plans we reviewed.

Planners often face the problem of setting priorities for action in a comprehensive plan. Typically, a traditional stand-alone comprehensive plan will contain several goals and dozens of recommended actions without a sense of the relative importance of each goal or action [14]. This problem becomes even more difficult when a constellation of plans exists, requiring coordination both among the subplans and between the subplans and the central comprehensive plan. A good practice is to use a theme approach to the central comprehensive plan, such as Tucson's 'Environments,' and then set priorities for goals and actions by theme. Some comprehensive plans break out actions in terms of cost and short-, medium-, and long-term actions. Additionally, as the survey of planners not surprisingly confirmed, the functional topics that receive funding are going to be the most implemented portions of the comprehensive plan.

There are important benefits to the constellation approach as well. While many have been identified throughout the article, three stand out:

(1) The constellation approach allows for more specific tailoring of human needs assessments through public participation on specialized topics.

(2) The constellation structure fosters timeliness and speed in addressing specific topics of concern in ever-changing urban environments. This is especially critical given the increased frequency of natural disasters and disease outbreaks worldwide.

(3) The constellation structure fosters collaboration between planning departments and different city departments or other agencies.

The legal structure in the United States requires an up-to-date comprehensive plan to support local ordinances, such as zoning, which implement planning law. Currently, federal policy and planning best-practices are pushing for reform of zoning, infrastructure design, and other planning regulations [11,13,15,27]. Goals for reform include improved equity, mobility, and climate resilience in cities. The United States is not alone in facing these challenges. The benefits that the constellation-type of comprehensive plan could provide appears evident, and many of the newer comprehensive plans are using the constellation structure. In short, the constellation-type of comprehensive plan is likely to become more common in the coming years.

## 5. Conclusions

We examined the comprehensive plans of 39 diverse US cities to identify cities that have a constellation-type of plan. We identified 20 cities whose comprehensive plans were structured as a central comprehensive plan tied to five or more functional, special topic, and neighborhood subplans. The central comprehensive plan contained a written connection to the subplans and a synthesis of the data, goals, and objectives of the subplans. Sixteen of the constellation plans were less than five years old, suggesting an emerging new trend in comprehensive planning. Although US cities widely adopted the single, stand-alone comprehensive plan during the 20th century, the constellation-type of comprehensive plan appears to be gaining popularity in the 21st century. Our brief survey of city plans in other countries suggests that the constellation structure of city plans may be gaining traction, at least in English-speaking countries.

We rated 18 of the US city 20 comprehensive plans excellent or proficient according to their Plan Connections, Plan Consistency, and Plan Bridging Gaps in their subplans and the central comprehensive plan. We received questionnaire responses from 14 cities that are using the constellation of plans approach. Planners noted that transportation and housing were the most common subplans. They also pointed out the ease of adding topics and updating subplans and the central comprehensive plan with the constellation structure. The greater amount of data can also aid in strengthening applications for grant funding from regional, state, and federal agencies. Subplans tailored to high-need areas, such as transportation and housing, will build the argument for why the city merits the funding and how the city will use the money. Planners mentioned that funding was the biggest obstacle to planning success. The potential strength of the constellation-type plan to provide a greater amount of data to bolster applications for grant funds is notable and important.

Future research on a broader set of cities, both in the US and internationally, would be useful to corroborate or modify our findings. The performance of the constellation of plans over time bears watching and analysis. However, it is important to note that many US cities that follow the single comprehensive plan model have allowed their comprehensive plans to become more than 10 years old, and are thus outdated.

The constellation of plans structure suggests a continuous planning effort with greater potential to increase the timeliness and responsiveness of the comprehensive plan to a city's changing needs than the traditional stand-alone comprehensive plan. Digitized subplans and maps mean that planners can more readily analyze and update data to keep a comprehensive plan current, such as through a review every 3 to 5 years, rather than an entire re-write of a plan every 10 years. In addition, responses from planners

suggest that the constellation structure may provide more opportunities to add subplans to provide greater depth and more topics to the comprehensive plan. Updating different neighborhood subplans at different times is a common practice mentioned by the planners. The need for continuous comprehensive planning and frequent updates may also increase the importance of having more planners on staff for each city, rather than relying on outside consultants.

**Author Contributions:** Data curation, C.Q.; Formal analysis, T.D.; Investigation, C.Q.; Methodology, C.Q.; Writing—review and editing, T.D. All authors have read and agreed to the published version of the manuscript.

**Funding:** This research received no external funding.

**Institutional Review Board Statement:** Not applicable.

**Informed Consent Statement:** Not applicable.

**Data Availability Statement:** Data on comprehensive plans are avaiable from the individual cities we surveyed.

**Conflicts of Interest:** The authors declare no conflict of interest.

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
