# Peer review of "The Constellation of Plans: Toward a New Structure of Comprehensive Plans in US Cities"

_land, doi:10.3390/land11101767_

Round 1

Reviewer 1 Report (Previous Reviewer 2)

The paper is fine and can be published

Author Response

Reviewer #1 wrote:

The paper is fine and can be published.

THANK YOU. WE APPRECIATE YOUR EXTENSIVE COMMENTS ON THE EARLIER DRAFT.

Reviewer 2 Report (Previous Reviewer 3)

The content is fine but there are lots of sloppy errors which I pointed out before, but have not been rectified.

For example, lines 16-18: 'The responses of planners s illustrate how this approach to comprehensive planning allows cities to increasing size and number of functional and area subplans' - this makes no sense.

line 30 - closing square bracket is missing [1, 7,8; 

line 41 - references in both round and square brackets ([1-20] - this is just one example - there are others

line 253 onwards - line spacing is different

Table 3 has formatting issues

reference 8 - Author 1 - what is this?

reference 22 - Author 2 - ditto

Author Response

See attached file.

This manuscript is a resubmission of an earlier submission. The following is a list of the peer review reports and author responses from that submission.

Round 1

Reviewer 1 Report

Dear Authors,

Here you can find some important recommendations regarding a eventual publication of your study. I wish you good luck for your work.

Comments

A clear explanation of Figure 1 is missing.

Also, authors have mentioned that “(…) the difference between the two plan structures that depends on: 1) the number of subplans tied to the central comprehensive plan; and 2) how well the subplans are connected to and inform the comprehensive plan, and vice versa”. Even considering the explanation given on the differences between both plans (lines 69-75), it would be interesting to have also an explanation in terms of their tendence and time evolution (albeit in a systematized way).

Lines 148-151 – Authors wrote: “We suggest that a good quality constellation-style of comprehensive plan offers several advantages over the traditional comprehensive plan”. It is very strange that the authors are making suggestions to the readers at this point in the text, even before presenting and discussing the results of the study. And, the considerations made on the constellation-type comprehensive plan (lines 151-164) are based on what? On the literature review? On authors’ previous studies/publications? On this present study? The authors venture considerations that have little or no scientific basis (so it seems, since it is not clear where these considerations are based on).

In fact, although the authors refer, theoretically, to the advantages of the constellation of plans structure, it is never properly understood how these advantages really become effective in the planning practice and what is the degree of efficiency of these model plans. Especially for those who do not live or know the reality of the North American planning system, the explanations provided, without the illustration of some practical examples, leave very vague reasons. This article lacks this illustrative power, which I strongly recommend in advance.

Somehow, the Discussion section and the Conclusions section could merge, as some of the central ideas are repeated.

Reviewer 2 Report

The paper is of interest for the journal because it provides an understanding of whether US cities are structuring comprehensive plans to resemble a ‘constellation’ of functional, special topic, and neighborhood subplans tied to a central guiding plan. This is why this paper contributes to update the field of urban studies. Tables and Figures are OK and I am quite satisfied with the paper.

Nevertheless, I highly suggest to add the following aspect, i.e. the urgency for plans to foster resilience and sustainability (cite for instance https://journals.sagepub.com/doi/10.1177/00420980211045571).

Between lines 133 and 147 (or just before these lines) I suggest to add the international dimension of the urgent change of approach in planning to provide a more human-centered environment, for instance through specific urban mobility plans (cite for instance this reference, https://link.springer.com/chapter/10.1007/978-3-031-10542-5_18) or including specific actions into planning, such as Superblocks and the 15-minute city (cite for instance these references: https://www.nature.com/articles/s41893-022-00855-2,  https://www.nature.com/articles/s41599-022-01145-0). After doing this, please, vriefly add some considerations for the US cities in the discussion and conclusion sections. Discuss how an approach towards a more human needs satisfaction is changing urban plans in the US and define also futher research needed towards a better understanding of this aspect in the US but also abroad.

This is why I require a new version of the paper before its acceptance.

Reviewer 3 Report

This is a very interesting paper on recent developments in the US planning system. On the whole it is well written although the proofs will need to be read carefully as there are a few typos. Why are the in text references in both square brackets and parentheses? This is unusual. Table 3 and the references both have some formatting issues.